# Asymmetry of carbon sequestrations by plant and soil after forestation regulated by soil nitrogen

Songbai Hong[1], Jinzhi Ding [2] ✉, Fei Kan[1], Hao Xu[1], Shaoyuan Chen[1], Yitong Yao[3] & Shilong Piao [1,2] ✉

Forestation is regarded as an effective strategy for increasing terrestrial carbon sequestration. However, its carbon sink potential remains uncertain due to the scarcity of large-scale sampling data and limited knowledge of the linkage between plant and soil C dynamics. Here, we conduct a large-scale survey of 163 control plots and 614 forested plots involving 25304 trees and 11700 soil samples in northern China to fill this knowledge gap. We find that forestation in northern China contributes a significant carbon sink (913.19 ± 47.58 Tg C), 74% of which is stored in biomass and 26% in soil organic carbon. Further analysis reveals that the biomass carbon sink increases initially but then decreases as soil nitrogen increases, while soil organic carbon significantly decreases in nitrogen-rich soils. These results highlight the importance of incorporating plant and soil interactions, modulated by nitrogen supply in the calculation and modelling of current and future carbon sink potential.

Forestation (including both afforestation and reforestation) is generally advocated as an effective method to increasing ecosystem carbon sequestartion[1]. Over the past few decades, forestation has been widely implemented on a global scale, resulting in a substantial expansion of the planted forest area[2]. Large-scale forestation has been widely reported to improve some ecosystem services, e.g., increasing wood production, soil and water conservation, and enhancing carbon sequestration[3–7]. For instance, forestation in China has contributed substantially to global greening[8] and the recent reversal of the loss of global terrestrial biomass[9]. However, as one important aspect of ecological consequence of forestation which has attracted broad attention, its carbon sequestration potential remains highly uncertain and controversial[1,10–13]. Large-scale field investigations have been scarce, limiting a comprehensive understanding of carbon dynamics after forestation, particularly the interactions between plant biomass and soil carbon dynamics[13,14]. Therefore, a holistic evaluation of the carbon sequestration induced by forestation at a large scale, which provides

foresight and guidelines for the future implementation of forestation projects, is challenging but extremely valuable.

Forestation has the potential to increase carbon sequestration by expanding both plant biomass and soil carbon stock. The increase in biomass carbon storage is a major component of the forestation-induced carbon sink[6,13]. In similar climate conditions, forests generally hold a larger biomass carbon stock than other ecosystems[15], and it has been estimated that the global potential for increased carbon storage in biomass is more than three times that of soil[13]. In addition to biomass, the second largest carbon sink is soil in the planted forest ecosystem[6,13]. Although the response of soil carbon to forestation may be slower than the response of biomass, the soil can still strongly regulate the ecosystem carbon balance due to the large carbon stock[16]. In most studies[9,17,18], a fixed ratio between biomass and soil organic carbon (SOC) has generally been used to estimate SOC and ecosystem total carbon stocks based on the strong association between plant biomass and SOC stock under an assumption of proportional carbon

[1]Institute of Carbon Neutrality, Sino-French Institute for Earth System Science, College of Urban and Environmental Sciences, Peking University, 100871 Beijing, China. [2]State Key Laboratory of Tibetan Plateau Earth System, Environment and Resources (TPESER), Institute of Tibetan Plateau Research, Chinese Academy of Sciences, 100101 Beijing, China. [3]Division of Geological and Planetary Sciences, California Institute of Technology, Pasadena, CA 91125, USA. ✉e-mail: jzding@itpcas.ac.cn; slpiao@pku.edu.cn

dynamics of biomass and SOC. However, an increase in biomass does not necessarily enhance SOC accumulation, and may even result in a SOC loss, as has been observed under $CO_2$ fertilization[19–21] and in warming systems[22]. The asymmetry of biomass and soil carbon dynamics may be regulated by climate and soil nutrient supply[23], and, in particular, nitrogen[20]. Unfortunately, this mismatch of carbon dynamics between biomass and SOC after forestation has not yet been thoroughly investigated, hindering our ability to make comprehensive understanding of carbon dynamics after forestation and accurate predictions of the carbon sink potential of forestation.

Here, we conducted a comprehensive, large-scale, paired-survey investigation of the impacts of forestation on biomass and soil carbon stock in northern China. Large-scale afforestation and reforestation programs have been implemented in China since the 1970s[24], and currently China has the largest area of planted forests in the world[2,25]. The Three-North Shelterbelt Development Program (TNSDP), situated in the northeastern, northern and northwestern regions of China, represents the earliest and the most extensive forestation initiative in China[26,27]. The project covers more than 4,000,000 km², and is spread across large gradients of climate and soil conditions, providing an ideal opportunity to explore the ecological impacts of forestation in different environments. The pairwise survey campaign was conducted within TNSDP area during 2012 and 2013, and involved 163 control plots and 614 forested plots (Supplementary Fig. 1), with 25,304 trees and 11,700 soil samples being surveyed (see Methods). By using this extensive field dataset, we estimated the carbon sink due to forestation in northern China via a machine learning approach. Furthermore, we compared the dynamics of biomass and SOC after forestation along climate and soil nitrogen gradients to test whether the changes of biomass and SOC were symmetrical (larger biomass means more carbon input to soil) or asymmetrical (plants nutrient acquisition accelerates decomposition of soil organic matter).

## Results
### Changes in ecosystem carbon sequestration with forestation
Across all the control-forestation plot pairs, the average change of biomass density induced by forestation (Δ(biomass density) = biomass density in forested plot – biomass density in control plot) was 4.80 kg C m⁻², indicating a significant ($P < 0.001$) increase of biomass density after forestation. Δ(biomass density) increased with stand age, with the rate of increase varying across tree species (Fig. 1). Specifically, the largest increases of Δ(biomass density) with stand age were found for *Larix* (*L.*) *gmelinii* (Fig. 1b) and *Pinus* (*P.*) *sylvestris* var. *mongholica* (Fig. 1c), indicating the fast growth rate of these two tree species. Consequently, forestation with *P. sylvestris* var. *mongholica* resulted in the largest increase of biomass density (6.89 kg C m⁻²), followed by *L. gmelinii* (5.41 kg C m⁻²) (Fig. 1f). In contrast, the smallest average Δ(biomass density) was observed for *P. tabuliformis* (1.40 kg C m⁻²) (Fig. 1f), which had only a small increase with stand age (Fig. 1d). Δ(biomass density) for *Populus* spp. increased slightly with stand age (Fig. 1e), with an average value of 4.92 kg C m⁻². For *P. koraiensis*, the slope between Δ(biomass density) and stand age was 0.20 kg C m⁻² yr⁻¹ (Fig. 1a), and its average Δ(biomass density) was 4.38 kg C m⁻² (Fig. 1f).

The biomass carbon sequestration of forestation also varied among original vegetation and land use type (Supplementary Table 1). The largest carbon sequestration was observed for forestation on natural forest and cropland, especially when forested with *P. sylvestris* var. *mongholica* on them. By contrast, the smallest carbon sequestration was observed for forestation on grassland.

To estimate forestation-induced biomass carbon sequestration for the whole study region, a model tree ensemble (MTE) approach was employed (see Methods). Mean annual precipitation (MAP), mean annual temperature (MAT), tree species, stand age, longitude, and latitude were used as the predictors in the MTE. The good performance of this approach (Supplementary Fig. 2) allowed us to generate

the distribution of Δ(biomass density) for the whole forestation region of northern China. As shown in Fig. 1g, Δ(biomass density) exhibited high spatial heterogeneity, with a large increase in biomass density in the east of the study region (Jilin province), where large values of planted forest biomass density were measured during the field survey (Supplementary Fig. 1). In the south of the study region, we generally found relatively smaller increases in biomass density induced by forestation, and with some scattered areas (16%) showing decreased biomass density (Fig. 1g).

Based on the planted area for each tree species (Supplementary Table 2, see Methods), forestation in northern China increased biomass carbon stock by 678.25 ± 37.98 Tg C (Table 1). Combining with the increase of SOC (234.94 ± 9.6 Tg C), forestation in northern China increased the total organic carbon (TOC = biomass C + SOC) stock by 913.19 ± 47.58 Tg C. When averaged over the planted area, forestation increased TOC density by 7.42 ± 0.39 kg C m⁻², with biomass and SOC contributing 5.51 ± 0.31 kg C m⁻² (accounting for 74%) and 1.91 ± 0.08 kg C m⁻² (accounting for 26%), respectively.

The forestation-induced carbon sequestration varied strongly across planted tree species due to both divergent changes in carbon density and different planted area (Table 1, Supplementary Table 2). The largest contribution came from *L. gmelinii*, which increased biomass carbon stock by 236.97 ± 13.93 Tg C, and increased TOC stock by 346.84 ± 15.73 Tg C. *Populus* spp., which had the largest planted area in northern China, increased the biomass carbon and TOC stocks by 227.36 ± 7.93 Tg C and 287.25 ± 12.23 Tg C, respectively. The group of 'other' tree species, was another important contributor of increased TOC stock, accounting for 200.22 ± 13.24 Tg C, with 168.05 ± 11.30 Tg C from biomass. Forestation with *P. koraiensis* and *P. sylvestris* var. *mongholica* increased biomass carbon stocks by only 22.29 ± 1.34 Tg C and 14.96 ± 1.95 Tg C, respectively, due to their small planted areas. *P. tabuliformis* only increased biomass carbon stocks by 8.61 ± 1.52 Tg C mainly because of its small increase in biomass density. The increase in TOC stocks induced by *P. koraiensis*, *P. sylvestris* var. *mongholica* and *P. tabuliformis* were 33.55 ± 1.51 Tg C, 19.01 ± 2.16 Tg C and 26.31 ± 2.10 Tg C, respectively.

### Divergent responses of biomass and soil organic carbon to forestation
Although forestation increased both biomass and soil stocks, the changes in biomass and SOC were not necessarily synchronous. The ratio of Δ(biomass density) and ΔSOCD had large variation and spatial heterogeneity (Fig. 1h). More surprisingly, biomass and SOC showed contrasting responses to forestation in 29% of pixels. To further explore which environmental factors drive the associations between the biomass and SOC stock in forested ecosystems, we compared the changes in biomass and SOC induced by forestation along the climate and soil nitrogen gradients (Fig. 2). Large increases in biomass density were mostly found in areas with MAT lower than 4 °C (Fig. 2a). In contrast, negative changes in biomass density were found in areas with MAT higher than 8 °C. However, we did not observe similar patterns for the responses of SOC to forestation in the same climate space (Fig. 2b), which implies that the increase (decrease) of plant biomass does not necessarily induce the increase (decrease) of SOC. Combining biomass and SOC, we found that relatively larger TOCD increases were generally distributed in areas with lower MAT (< 6 °C) (Fig. 2c). However, smaller TOCD increases (even decreases) often occurred in warmer areas (MAT > 8 °C) (Fig. 2c), mainly because of the compensation between biomass and soil C changes (Fig. 2d). Consistent results were observed when aridity index was used in place of MAP (Supplementary Fig. 3).

Soil nitrogen strongly regulates carbon dynamics of both plant and soil[20, 23], therefore we also investigated how Δ(biomass density) and ΔSOCD varied with soil total nitrogen density (STND) and stand age (Fig. 2e–h). As shown in Fig. 2e, the larger increases of biomass

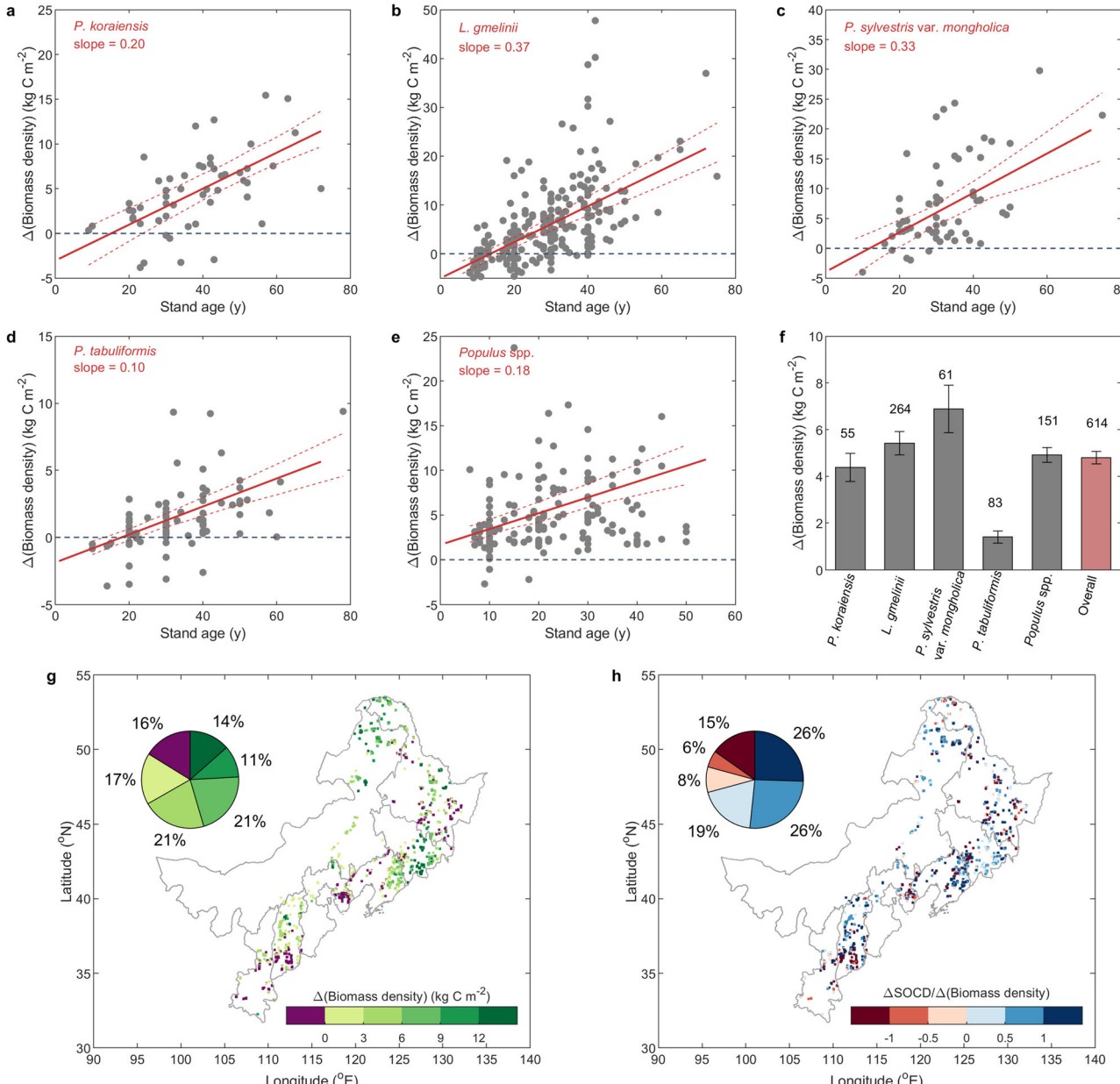

**Fig. 1 | Changes in biomass density (Δ(biomass density)) induced by foresta-tion. a–e** Relationships between Δ(biomass density) and stand age for *P. koraiensis, L. gmelinii, P. sylvestris* var. *mongholica, P. tabuliformis* and *Populus* spp. The solid lines in (**a–e**) are the results of linear mixed models, while the dashed lines mark the 95% confidence interval of the regressions. **f** The averaged values of Δ(biomass density) for different planted tree species and overall. Error bars indicate standard errors. The numbers on top of the bars indicate the sample size of each group. **g** The spatial distribution of Δ(biomass density) derived from upscaling via model tree ensemble (MTE). **h** The spatial distribution of the ratio of change in soil organic carbon (ΔSOCD) to Δ(biomass density). The resolution of the data in (**g** and **h**) is 1 km. The inset pie chart shows the percentage of pixels in each group. The base maps in (**g** and **h**) were derived without endorsement from GADM data (https://gadm.org/), and the maps were generated in MATLAB R2020a (MathWorks).

density induced by forestation were found in areas rich in soil nitrogen and with older stands. In contrast, where soil nitrogen levels were poor and the forest was young, forestation only resulted in a small increase in biomass density, and even a decrease in some areas. The change in SOCD was much smaller than that of biomass density, with different variations along soil nitrogen gradient (Fig. 2f). Specifically, forestation generally decreased SOCD in soils with high nitrogen content, while it increased SOCD in soils with low nitrogen content, especially in older stands. Interestingly, when STND was in the range of 1.0–1.2 kg N m⁻² and stand age was more than 40 years, we found that both biomass density and SOCD showed large increases (Fig. 2g). These areas, therefore, had the most pronounced increase of TOCD, indicating that

forestation in areas with moderate soil nitrogen can contribute to large values of carbon sequestration after the growth of planted trees.

The foregoing results, based on the outputs of machine learning, indicate the divergent variations between ΔSOCD and Δ(biomass density) along the soil nitrogen gradient, which is further confirmed by the results of field sampling data (Fig. 3). With the increase of STND, Δ(biomass density) increased initially but then gradually decreased, while ΔSOCD decreased monotonically (Fig. 3a). The combination of these two trends induced nonlinear changes of ΔTOCD along soil nitrogen gradient (Fig. 3b). When STND was less than 1.5 kg N m⁻², forestation significantly increased TOCD, and the magnitude increased with STND (Fig. 3c). The largest carbon sequestration due to

**Table 1 | The upscaling estimation of carbon sequestration induced by forestation**

| Change in C density (kg C m$^{-2}$) | | | |
|---|---|---|---|
| | Δ(Biomass density) | ΔSOCD | ΔTOCD |
| P. koraiensis | 8.98 (0.53) | 4.20 (0.09) | 13.18 (0.62) |
| L. gmelinii | 7.54 (0.31) | 3.78 (0.08) | 11.32 (0.39) |
| P. sylvestris var. mongholica | 3.25 (0.52) | 0.95 (0.28) | 4.20 (0.80) |
| P. tabuliformis | 0.54 (0.10) | 1.28 (0.05) | 1.82 (0.15) |
| Populus spp. | 4.86 (0.20) | 0.73 (0.11) | 5.59 (0.31) |
| Other | 6.07 (0.30) | 0.91 (0.03) | 6.98 (0.33) |
| Average | 5.51 (0.31) | 1.91 (0.08) | 7.42 (0.39) |
| **Change in Total C (Tg C)** | | | |
| | Biomass | SOC | TOC |
| P. koraiensis | 22.29 (1.34) | 11.26 (0.17) | 33.55 (1.51) |
| L. gmelinii | 236.97 (13.93) | 109.87 (1.80) | 346.84 (15.73) |
| P. sylvestris var. mongholica | 14.96 (1.95) | 4.05 (0.21) | 19.01 (2.16) |
| P. tabuliformis | 8.61 (1.52) | 17.70 (0.58) | 26.31 (2.10) |
| Populus spp. | 227.36 (7.93) | 59.89 (4.30) | 287.25 (12.23) |
| Other | 168.05 (11.30) | 32.17 (1.94) | 200.22 (13.24) |
| Total | 678.25 (37.98) | 234.94 (9.60) | 913.19 (47.58) |

Numbers in the brackets indicate the standard errors for multiple simulations. Note that the average changes in C density were calculated using the total C change divided by the area of planted forests. Data of soil organic carbon (SOC) are based on ref. 36.

forestation was observed in the group with STND in the range 1–1.5 kg N m$^{-2}$. When STND exceeded 1.5 kg N m$^{-2}$, although forestation increased the carbon density in biomass, the loss of SOC offset the increase, so that any changes in TOCD were weak and nonsignificant.

Interestingly, we found a trade-off between SOC and biomass carbon dynamics among different planted tree species (Fig. 3d). For the species with a fast increase of Δ(biomass density) (e.g., *L. gmelinii* and *P. sylvestris* var. *mongholica*), ΔSOCD showed a large decrease along the soil nitrogen gradient. In contrast, for species with a slow increase of Δ(biomass density) (e.g., *P. tabuliformis*), ΔSOCD only showed a small decrease with increasing STND. These results further suggest that increasing plant biomass may induce soil carbon loss due to nutrient acquisition.

## Discussion

This study provides a comprehensive assessment of forestation-induced carbon sequestration through a large-scale pairwise field investigation involving both plant biomass and SOC. Using strict paired design and widespread survey of 25,304 trees and 11,700 soil samples, our study found that forestation in northern China over the past few decades has contributed to a large carbon sink of 913.19 Tg C, with 74% coming from the biomass and 26% from SOC. From the first implementation of TNSDP in 1978 to the sampling year 2012, the mean annual carbon sequestration due to forestation in northern China has reached 26.86 Tg C year$^{-1}$. Such a value equates to 10.3% of the terrestrial carbon sink in China (0.26 Pg C year$^{-1}$, based on ref. 25) and 14.9–19.2% of the forest carbon sink in China (0.14–0.18 Pg C year$^{-1}$, based on ref. 18). These results suggest that forestation can create a considerable carbon sink and is a potential solution to the problem of increasing atmospheric $CO_2$ concentrations.

Our results further indicate that the forestation-induced carbon sink is regulated by many factors, which can provide benchmarks for future forestation. First, we observe that the carbon sequestration capacity varies greatly among tree species, which highlights the importance of the appropriate species choice for forestation.

Moreover, multispecies plantations have been found to hold a larger biomass than monocultures[28], so the effects of biodiversity and tree species composition should also be carefully evaluated to maximize carbon sink benefits[29]. Second, local climate and soil conditions also strongly affect the forestation carbon sink. Specifically, forestation in humid (cold) regions sequesters carbon more effectively than in dry (warm) regions (Fig. 2, Supplementary Fig. 3). At the same time, our results reveal that the carbon sink would be larger in regions with moderate nitrogen levels than in regions with high nitrogen levels because of the large carbon loss in nitrogen-rich soils. Therefore, careful and evidence-based site selection and nutrient management are also necessary during forestation campaigns. Third, forest biomass density generally increases rapidly with stand age, especially for newly-planted forests, thus it is reasonable to expect that China's planted forest will continue to provide a carbon sink in the coming decades[30]. Indeed, it is estimated that age-related forest biomass C sequestration in China will be 6.69 Pg C (about 0.17 Pg C year$^{-1}$) from the 2000s to the 2040s, and will be enhanced by ongoing climate change and increasing $CO_2$ concentration[31]. However, it should be recognized that the carbon sequestration capacity of forest will gradually saturate as stand age increases[31,32], so appropriate forestation timing and scientific forest management are also valuable choices. Furthermore, forestation is not panacea to climate change and carbon sequestration is just one of many ecological and social aspects of forestation[33]. It has been found that simple and unreasonable forestation may also bring both environment and social problems[34,35]. Therefore, collectively, a sustainable forestation carbon sink requires integrated planning and scientific decision making (to plant or not to plant), accounting for local climates and conditions (where to plant), temporal dynamics of forestation-induced carbon sink (when to plant) and careful species choice and management (how to plant), which are all extremely important if China is to achieve its 'carbon neutrality' target.

It is noteworthy that some uncertainties may be involved in our study. For instance, although original vegetation and land use type was found to affect carbon sequestration of forestation, we did not include this variable in our MTE model due to the lack of reliable regional-scale data. However, we conducted further analysis by training MTE models with and without previous land use type involved and found model with vegetation and land use type did not show much better performance (Supplementary Fig. 4), indicating the MTE model used is sufficiently reliable to estimate biomass carbon sink of forestation. In addition, the resolution of data for control groups (300 m) is larger than the size of our sampling plots, which may also bring some uncertainties. Nevertheless, it is acceptable because we have taken efficient efforts to make our sampling plots as representative as possible for the local environment, which can minimize the uncertainties brought by the mismatch in the spatial resolution.

In previous studies on the large-scale carbon balance, SOC stocks were generally estimated as the product of a fixed ratio and biomass carbon storage, due to the scarcity of large-scale SOC measurements[9,18]. Our pairwise sampling system provides a more accurate understanding of soil carbon dynamics after forestation at the regional scale[36], which allows us to further investigate the carbon-dynamics relationships between soil and biomass. Our results indicate that the assumption of a constant ratio between SOC and biomass carbon is unreliable. The reasons for this unreliability may be threefold.

First, changes in biomass and SOC after forestation are asynchronous. In theory, both biomass and SOC would have nonlinear responses to forestation[30,37,38]. Forest biomass would initially increase with stand age but after some time would reach a relative equilibrium[30,38]. Here, we found that forestation generally increased biomass carbon density but the equilibrium state was not reached. This may be due to the young forest age and the large spatial heterogeneity among the study sites. Soil carbon dynamics after forestation

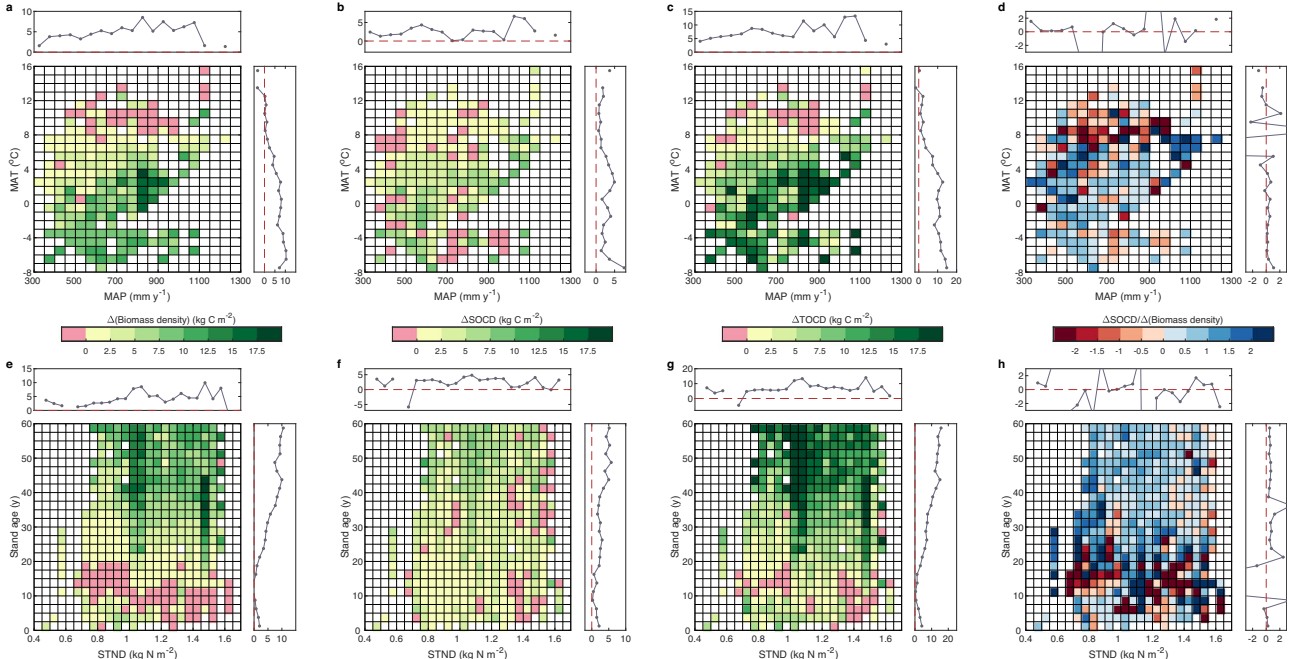

**Fig. 2 | Divergent responses of biomass and soil organic carbon densities to forestation (Δ(biomass density) and ΔSOCD) along climate and soil nitrogen gradients. a–d** The distribution of Δ(biomass density), ΔSOCD, ΔTOCD (i.e., Δ(biomass density)+ΔSOCD), and ΔSOCD/Δ(biomass density) in a two-dimension space of mean annual temperature (MAT) and precipitation (MAP). **e–h** The distribution of Δ(biomass density), ΔSOCD, ΔTOCD, and ΔSOCD/Δ(biomass density) in a two-dimension space of soil total nitrogen density (STND) and stand age. The mean values for each interval, derived from the output of machine learning, are shown. The top line chart in (**a**) indicates the variation of Δ(biomass density) along the MAP gradient, while the chart on the right-hand side indicates the variation of Δ(biomass density) with MAT. Mean values for each interval were used to generate the lines. The line charts in (**b–h**) were created similarly.

is ultimately determined by the balance between carbon input and decomposition[37,39-41], thus it is not necessarily synchronous with changes in biomass carbon. Indeed, forestation-induced change in SOCD was mostly determined by the initial SOC storage, and the magnitude of change gradually decreased with stand age[36].

Second, the association between biomass density and SOCD changes along the soil nitrogen gradient, implies that the above- and below-ground interactions are regulated by plant nutrient acquisition. In general, a larger biomass can produce more litter and increase carbon input to the soil carbon pool. Thus, a positive relationship between biomass and SOC is widely used in most terrestrial ecosystem models[14,42]. However, the increase of plant biomass requires a larger nutrient supply, and can stimulate the decomposition of soil organic matter to obtain more nitrogen[43]. In areas with a large amount of soil organic matter, the input of litter to the SOC pool cannot completely recharge the strong decomposition of SOC (may be due to the priming effect)[44], and hence we observed a large decrease of SOC. Such phenomenon was also observed in tundra, where high plant activity during the growing season stimulates the decomposition of soil organic matter[45]. Long-term control experiments further found that doubled aboveground litter additions did not increase soil C for any of the forests studied likely due to long-term soil priming[46]. Changes in soil nitrogen after forestation along the nitrogen gradient support this mechanism (Supplementary Fig. 5), where we found significant decrease of soil nitrogen after forestation in areas rich in soil organic matter[47]. The combination of increasing carbon input to SOC due to the increase of litter and the increasing decomposition of SOC due to nutrient acquisition after forestation complicates SOC dynamics, and requires further investigation.

Third, the ratio between biomass carbon sequestration and soil carbon sequestration varied greatly among tree species. For instance, the largest ratio between biomass and soil carbon sequestration was observed for 'other' group (about 5.2), while the smallest ratio (about 0.5) was observed for *P. tabuliformis*. Such differences may be caused

by the divergent strategies for carbon allocation and nitrogen acquisition among species[48]. Indeed, we found that tree species with a faster increase in biomass density were generally accompanied by a larger SOC loss along the soil nitrogen gradient (Fig. 3d), implying that the nutrient acquisition strategy is species-specific and strongly associated with plant biomass[49].

Forestation regulates the dynamics of SOC via affecting both C inputs and outputs[50,51]. Besides the litter input and nutrient acquisition (e.g., priming effect), forestation could also regulate SOC dynamics indirectly via changing soil biological or microclimatic conditions (soil temperature, moisture, pH and et.)[52,53]. Moreover, disturbance and forest management can also affect the input and output of SOC[54-56]. These effects make dynamics of SOC after forestation more complicated and leave the interaction between biomass and soil C cycles more uncertain.

At present, Earth system models (ESMs) generally produce a strong response of SOC to an increase in C input (i.e., NPP)[14,42]. However, our results suggest that the dynamics of plant C and SOC are not proportionally synergistic and there may be a trade-off between them due to nutrient competition. Such nutrient competition mechanisms have also been observed in tundra forest[45], and further confirmed by a synthesized study focusing on $CO_2$ fertilization[20]. Moreover, both modeling[42] and field experiment[57] studies have found that increasing N input can enhance soil C sequestration. Given that ESMs are currently inadequate for modeling C and N couples[58], a better parameterization and description of C and N interaction schemes is likely to reduce model uncertainties in SOC dynamic simulation. Integrated studies, combining data from manipulative field experiments and large-scale sampling of soil C and N dynamics may yield some new insights, but we suggest that ecological theories of nutrient acquisition can also help to develop and refine the soil C-N schemes used in ESMs.

In summary, via an extensive paired investigation, we estimate that forestation in northern China has contributed a considerable carbon sink of 913.19 ± 47.58 Tg C, 74% of which is from biomass and

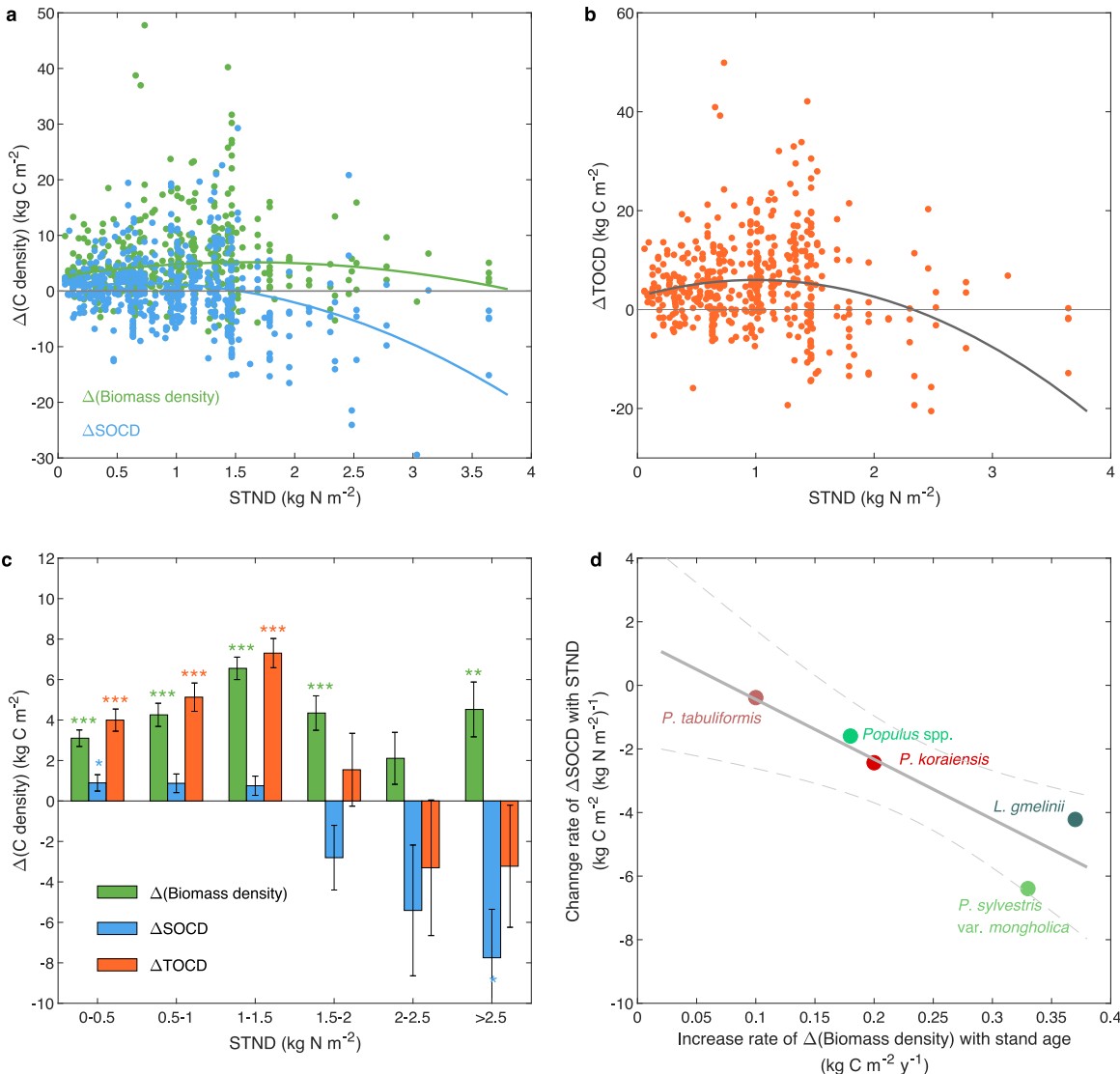

**Fig. 3 | Dependency of changes in carbon densities induced by forestation on background soil total nitrogen density (STND). a** Relationship between changes in carbon densities of biomass and soil organic carbon (SOC) with STND. **b** Relationship between changes in total organic carbon density (TOCD) and STND. Lines in (**a** and **b**) were fitted based on linear mixed model. **c** Comparison of changes in carbon densities in groups with different STND values. Independent sample *t*-tests were conducted, incorporating correction for false discovery rates, to compare the data of each group against a null hypothesis of 0. *, ** and *** indicate that the null hypothesis can be rejected at *p* < 0.05, 0.01 and 0.001, respectively. Error bars indicate standard errors. This figure is based on field sampling data at plot-level. **d** Trade-off between biomass and soil carbon dynamics among tree species. Increase rate of Δ(Biomass density) with stand age refers to the slope between Δ(Biomass density) and stand age (see Fig. 1a–e). Change rate of ΔSOCD with STND refers to the regression slope between ΔSOCD and STND.

26% from SOC. This forestation-induced carbon sink is regulated by many factors, e.g., tree species choice, local climate, soil nutrient supply and stand age, which can provide benchmarks for future forestation on where, when and how to plant. Moreover, we found that biomass and SOC showed divergent responses to forestation along the soil nitrogen gradient, implying a trade-off between biomass and SOC relating to plant nitrogen acquisition. Our results indicate that the wide use of a fixed ratio between biomass and SOC may overestimate the carbon sink potential of forestation, and that the assumption made in most ESMs, that biomass is positively associated with SOC, is not reliable. Therefore, we argue that more experiment results and ecological theories should be used to improve C-N schemes in ESMs.

## Methods
### Study region
The study region (34.20°–51.80°N and 106.81°–133.31°E) covers seven provinces of northern China (i.e., Heilongjiang, Jilin, Liaoning, Hebei,

Shanxi, Shaanxi and the Inner Mongolia) (Supplementary Fig. 1) and has a diverse range of climatic conditions, with MAT ranging from about −8 °C to 16 °C and MAP from about 300 to 1300 mm. The United Nations Food and Agriculture Organization (FAO) soil classification system[59,60] lists the dominant soil types in the region as phaeozems, gleysols, humic cambisols, haplic/albic luvisols or eutric/dystric cambisols, haplic calcisols, kastanozems, chernozems, cambisols, haplic alisols, and ferric/haplic luvisols. Soil properties (e.g., pH, organic carbon, and nitrogen) present large gradients in the study area[36,52]. The region is a hot-spot of forestation in China, containing more than 120,000 km² of planted forests, most of which are attributed to TNSDP[24]. It is, therefore, an excellent region for investigating the ecological impacts of forestation under different climate and soil conditions.

### Field campaign
The control-forested pairwise system was primarily established to explore the effects of forestation on soil properties[36,52,61]. The system

consisted of 163 study sites, each of which contained a control plot (non-forested plot) and several forested plots (1–26) with different planted tree species and/or different stand age. The distance between any one forested plot and its corresponding non-forested-control plot was usually 50–100 m to minimize any differences in soil and climatic properties between the pair. The largest acceptable distance was 2.5 km, which applied in very few cases. Moreover, the original (pre-forestation) vegetation and land use type and soil type of each forested plot were the same as those of their control plots. The original vegetation and land use types consisted of barren land, cropland, grassland, natural forest and riparian sand land, so both afforestation and reforestation were considered in our study. Note that forestation on natural forests refers to the reforestation of previously clear-cut areas that were originally natural forests. Except cropland, other control groups (barren land, grassland, natural forest and riparian sand land) were not managed. Each control-forested pair, consisting of a forested plot and its corresponding control plot was utilized to provide a good assessment of the impact of forestation. It is also noteworthy that we do not rely on before-and-after comparisons since the control plot could not accurately represent the initial conditions. Instead, we focused on the differences in C stocks between forested versus non-forested areas, all else equal. This approach enables us to evaluate the cost/benefit of forestation accurately. In total, we surveyed 163 control plots and 614 forested plots.

All the forested plots are monoculture, with the dominant tree species being *P. koraiensis*, *L. gmelinii*, *P. sylvestris* var. *mongholica*, *P. tabuliformis* and *Populus* spp. (*Populus simonii*, *Populus beijingensis* and *Populus×xiaohei*). At each forested plot ($20 \times 20$ m²), we measured tree height (TH) and diameter at breast height (DBH) of each tree higher than 2.5 m, and used these data to calculate the biomass. Stand-age data were obtained from local forestry administrations, and were further validated by tree ring observations. In total, 25,304 trees were surveyed.

Soil samples were collected from both control and forested plots. At each plot, we dug three replicate soil profiles on the diagonal and collected samples from various layers (0–5, 5–10, 10–20, 20–30, 30–60 and 60–100 cm). We collected 18 samples in each plot (three replicates × six layers per replicate), except for a few plots which could not be cored to a depth of 1 m. In total, we dug 2331 soil profiles and obtained 11,700 soil samples. Note that we collected two cutting rings of soils at each depth, both of which were identical. One of rings was oven-dried while the other was air-dried in the laboratory. It should be noted that residues, litter, organic layers were not included during the soil sampling process.

### Biomass density and its forestation-induced changes

Biomass density in forested plots was calculated based on TH, DBH and the number of trees (NT). First, we calculated the timber volume ($V_i$) of each tree, based on Eq. (1).

$$V_i = a * DBH^e * TH^f \tag{1}$$

Coefficients ($a$, $e$ and $f$) for each tree species can be found in Supplementary Table 3. Equation (1) has been tested, and found to be appropriate, for the tree species in northern China in previous studies and the coefficients have been validated using local field survey data[62–66].

Second, we summed the timber volume of all trees in a plot:

$$V = \sum_{i=1}^{NT} V_i \tag{2}$$

and calculated biomass density ($B$) based on the timber volume (Eq. (3)). This equation is derived from the biomass expansion factor

method[67].

$$B = (bV + c)/A \tag{3}$$

where $A$ represents the plot area (400 m²), and values for $b$ and $c$ are given in Supplementary Table 4, which is based on ref. 67.

The change in biomass density induced by forestation was derived by subtracting the biomass density in the control plot from the biomass in the paired forested plot. Biomass density in control groups was extracted from the harmonizing vegetation-specific maps of both above and belowground biomass based on the longitude and latitude of each plot[68]. The datasets comprise both above and belowground biomass for different vegetation types (e.g., cropland, grassland, natural forest al.) at spatial resolution of 300 m in 2010, close to our sampling years (2012 and 2013).

### Laboratory work on soil properties

We oven-dried one cutting ring of soils to determine the soil dry weight (SDW) and bulk density. The other cutting ring of soils was air-dried to constant weight in a ventilated room. Roots and stones were removed by passing through 2 mm sieves and the soils were prepared for chemical analyses. Soil total carbon and soil total nitrogen content (STCC and STNC) were measured using an elemental analyzer (Viro el cube, Elementar, Germany). For soil inorganic carbon content (SICC), we used a 08.53 Calcimeter (M1.08.53.E, Eijkelkamp, Netherlands). SOC content (SOCC) was taken as the difference between STCC and SICC. Using SOCC and STNC, soil dry weight (SDW), volume of cutting ring ($V$), we were able to calculate the SOC and STN densities (SOCD and STND) of each layer in each plot (Eqs. (4) and (5)).

$$SOCD_j = SOCC_j * BD_j * Ps_j * w_j * 10^2 = SOCC_j * \frac{SDW_j}{V} * w_j * 10^2 \tag{4}$$

$$STND_j = STNC_j * BD_j * Ps_j * w_j * 10^2 = STNC_j * \frac{SDW_j}{V} * w_j * 10^2 \tag{5}$$

BD indicates soil bulk density, while Ps indicates the volume percentage of soil in each cutting ring (100% minus the volume percentage of roots and stones). In the equations, $j$ indicates the $j$th layer, and $w$ indicates the thickness of the layer, e.g., $j = 1$ indicate the first layer (0–5 cm, $w = 5$) while $j = 6$ indicates the sixth layer (60–100 cm, $w = 40$). Note that each plot had three replicate profiles, so mean $STND_j$ and $SOCD_j$ for the three profiles were used in the analysis. The sum of mean $STND_j$ and $SOCD_j$ in all layers was used to represent STN and SOC densities in each plot.

The difference of SOC between control and forested plots were used to indicate the impacts of forestation on SOC, which was then upscaled in the whole study region using boosted regression trees[36].

### Statistical analyses and upscaling estimation

In this study, data analyses were conducted based on control-forested pairs. The independent student's $t$ test was used to explore whether the change in biomass density (Δ(biomass density) = biomass density in forested plot - biomass density in control plot) induced by forestation was significantly different from 0. False discovery rates were corrected to control potential error rates in multiple comparisons[69]. Linear mixed models were performed to determine the relationships between carbon changes and stand age (Fig. 1) and STND (Fig. 3), where site number was added into the models as a random factor.

To upscale forestation-induced changes in biomass density, we applied an MTE approach. This method was first discussed by Jung et al[70]. and has since been widely used to upscale carbon (water) flux and storage to large spatial scales[71–73]. To avoid

overfitting, we randomly selected 80% of the sample data to form the training dataset, with the remaining 20% forming the validation dataset. The model trees were trained at the plot-level using MAP, MAT, tree species, stand age, longitude, and latitude as model inputs. Note that we used the change in biomass density induced by forestation as the dependent variable, so that we could estimate the carbon sequestration of forestation directly. The trained MTE is then applied to the validation dataset to test the performance of the MTE in simulating changes in biomass density. The output estimation is the median of 20 independent best ensemble members. We obtained changes in biomass density estimation in each 1 km spatial resolution grid cell by applying the well-trained MTE. The spatial distribution of planted forests at 1-km resolution was obtained from the report of the Seventh National Forest Resource Inventory (2004–2008), released by the State Forestry Administration of China[53]. Tree species data were derived from the 1:1,000,000 vegetation map of China[74]. MAP and MAT were obtained from the China Meteorological Forcing Dataset, which was created by merging a variety of data sources, and has a spatial resolution of $0.1 \times 0.1°$ and a temporal resolution of 3 h[75,76]. Stand-age data were derived from a forest age map of China[77], which was developed by downscaling the national forest inventory data to a 1 km spatial resolution. Data of aridity index were derived from global aridity index and potential evapotranspiration (ET0) database v3[78], which were used to validate the results based on MAP. In addition, data of soil carbon and nitrogen and biomass, and code related to this study have been deposited at figshare[79–81].

Importantly, the map of planted forests may not perfectly match the actual planted area for the study region, so the total regional change in biomass was calculated by weighting the simulated mean change in biomass density for tree group $j$ in province $i$ ($X_{ij}$) by its corresponding planted area ($A_{ij}$) (Eq. (6)). The planted area for each tree species in each province is shown in Supplementary Table 2. Note the value of $X$ for the 'other' tree species group was set to the mean value of the five major tree species.

$$\text{Total change in biomass} = \sum_{i=1}^{7} \sum_{j=1}^{6} X_{ij} A_{ij} \qquad (6)$$

Finally, we combined the estimated Δ(biomass density) and total change in biomass with changes in SOC density (i.e., ΔSOCD) and total change in SOC (see ref. 36 for SOC) to calculate the forestation-induced changes in total organic carbon density (i.e., TOCD) and stocks.

## Data availability
The field data of soil C and N related to this research have been deposited at figshare (https://doi.org/10.6084/m9.figshare.21341577). Data of biomass in planted forests and other related data have been deposited at figshare (https://doi.org/10.6084/m9.figshare.22346332). Data of aridity index are openly available at https://doi.org/10.6084/m9.figshare.7504448.v5. Data of biomass for control group are derived from https://doi.org/10.3334/ORNLDAAC/1763.

## Code availability
The code used in this research are available at figshare (https://doi.org/10.6084/m9.figshare.22346344).

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

## Acknowledgements

This study was supported by the National Key R&D Program of China (2019YFA0607304) and and the Chinese Academy of Sciences (CAS) Project for Young Scientists in Basic Research (YSBR-037). S.H. acknowledges the support from the Postdoctoral Innovation Talents Support Program of China (Grant No. BX2021005).

## Author contributions

S.P., S.H., and J.D. designed the study. S.H. conducted the experiments, performed the analysis and drafted the figures. S.H. and J.D. drafted the paper. S.H., J.D., F.K., H.X., S.C. Y.Y., and S.P. contributed to the interpretation of the results and to the text.

## Competing interests

The authors declare no competing interests.
