## [Peer Review File · Nature Communications]

Asymmetry of carbon sequestrations by plant and soil after forestation regulated by soil nitrogenReviewer #1 (Remarks to the Author):

The manuscript by Hong, Ding, and colleagues presents data on carbon stocks of trees and soils in plantations of northern China. They use a large data set (163 control sites and 614 tree plantations) to show that carbon sequestration depends on tree species, plantation age, and soil nitrogen. I believe that their data has potential to help clarify a limited, but important, set of questions about the carbon and nitrogen dynamics of plantations overtime. Unfortunately, this manuscript has several limitations that prevent it from improving our knowledge of how tree planting may contribute to Natural Climate Solutions, which seems to be the overarching aim stated in the abstract and introduction.

Firstly, I am concerned by the framing of afforestation as a "nature-based solution." The first sentence of the abstract references the seminar paper on Natural Climate Solutions by Griscom et al. (2018). But that paper focuses on reforestation NOT afforestation. A growing body of literature that demonstrates that afforestation via exotic plantations is problematic both socially and ecologically (see Fleischmann et al. 2020, *BioScience*) and will not help limit global warming to meet the Paris goals (see Lewis et al. 2019, *Nature*). These problems in China were also recognized by Cao et al. (2011, *Earth Science Reviews*), who point out that ecological inappropriate tree planting does more environmental damage than good. All of this is not to say that we should not study afforestation via monoculture tree plantations, but that the framing should clarify that carbon sequestration is just one of many ecological and social aspects to consider.

Secondly, the statistical methods overstate the level of replication. There 614 plantation plots but only 163 control plots, making it so that "several afforested plots (1-26)" (L 309-311) are associated with each control. The statistical methods make it appear that you consider the plantation-control pairs to be the unit of replication, with an $n=614$. This is incorrect because plantation plots located near each control plot are not independent. It would be more appropriate to use the $n=163$ sites (with 1 control plot per site) as the unit of replication. One solution is perform linear mixed models (rather than simple regressions) and to treat site as a random factor in linear.

Thirdly, the laboratory methods likely overestimate soil bulk density due to inclusion of water weight. Estimation of soil carbon stocks require accurate measurement of soil bulk density. It is standard to dry soils at 105 C to vaporize water and obtain a dry weight. Based on the methods (L355), soils were simply "air-dried to a constant weight in a ventilated room". Use of a non-standard method requires explanation.

Fourth, the afforestation sites included a wide range of previous land uses, which were not incorporated in models, and could have important implications for carbon dynamics and biodiversity. These included (L316-317), "barren land, cropland, grassland, natural forest, and riparian sand land." It is incorrect to call the clearing of natural forests with subsequent planting of monoculture tree plantations "afforestation".

Fifth, biomass was measured by the authors in the plantation plots, but not in the control plots. This is not only a methodological inconsistency, but assumptions about biomass in the control plots (and corresponding error) calls into question the ΔC estimates for the plantations. Specifically (L347-348): "Biomass density in cropland, barren land and riparian sand land was set to be 0." This is an underestimate of existing biomass (particularly below ground) on these sites. For "natural forest control sites" (L348-350) the authors did not measure trees, but rather used above ground biomass estimates from a remote sensing data set and then multiplied by a constant (1.24). Finally (L350-352), for grassland control sites (which certainly have the majority of biomass below ground) they use a MODIS (remote sensing) product that estimates net primary productivity. But this is a spatial mismatch, because the MODIS resolution is 500 m x 500 m whereas the control and plantation plots are 20 m x 20 m. Furthermore, there is insufficient description in the methods to understand how the MODIS estimates are used to calculate the grassland carbon stocks. All of this is concerning because the control plots are the basis for calculating ΔC biomass, which is key to the aims of the study.

In closing, I hope that the authors can use these critiques to revise and focus their analyses on a set of refined hypotheses for which their data is better suited. I trust that with such a comprehensive data set on plantation carbon stocks and Nitrogen that such a revision will focus on C and N dynamics over time, and less so on climate mitigation via C sequestration.

Reviewer #2 (Remarks to the Author):

This study uses a remarkable dataset where soil organic carbon and plant biomass have been assessed over 163 study sites of afforested and control plots over a large region in China. The study has several shortcomings that prevent me from making a positive recommendation. Here are my main concerns:

-Lack of information on the control treatment. Although an effort was made so that both treatments (afforested and control) shared the same initial conditions at the time of plantation, it is unclear how and whether the control plots were managed or not. This could have important implications for the interpretation of the results and this information is lacking. In particular, I understand that plantations of different ages were sampled once; i.e. there is no re-sampling as the plantations are aging. Therefore, the control may not reflect the initial conditions. This aspect needs to be clarified because the study is based on the assumption that the control represents plantation conditions at time zero.

-Methodology:

1- It is unclear whether residues, litter, or organic layers are included or not in the SOM assessment.

2- There is no mention that soil bulk density, a crucial element for estimating SOM stocks was assessed. There is also no information on the exclusion of coarse fragments (greater than 2mm). Although a cutting ring is mentioned in l.360; there is no mention of how rock fragments were dealt with.

-Statistical treatment:

There are on average six afforested plots that are compared to a single control plot. How was this considered in the statistical analysis? Also, the statement that there are 614 control-afforested plot pairs appears to be misleading as the number of pairs should be 163 (l. 309).

-Speculative assumptions:

1- In the manuscript, any loss of soil C is interpreted as being the product of priming. Soil organic matter priming is commonly defined as the modification of soil organic matter (SOM) decomposition by plant carbon (C) input. The authors present no evidence that changes in SOM are caused by plant carbon inputs. Plant carbon inputs have not been assessed. Other causes of SOM changes, including disturbance, management, and changes in soil biological or microclimatic conditions have not been assessed and are not discussed.

L.78; 207; 249; 254; 264; 272 (strong priming effect)

2- The authors have frequently referred to the assumption that many studies, assessments, and modeling use a direct relationship between biomass accumulation and changes in SOM. I do not believe that this is true, contrary to the authors' claim. Some model results would show such a correlation; but it is generally accepted that while plant biomass accumulates after afforestation, changes in SOM are frequently not apparent and can be negative or positive depending on several parameters related to previous land use and soil properties. The common understanding of the factors that are known to impact SOM accumulation with afforestation is not discussed. For example, a well-cited meta-analysis (Laganière et al. 2010) has identified major drivers of SOM accumulation following afforestation including land use and soil types. Also, the IPCC (2019), in its updated methodology based on an extensive literature review, does not refer to a relationship between productivity and SOM accumulation (IPCC). See also Mayer (2020).

Cited literature:

LAGANIÈRE, et al. (2010), Carbon accumulation in agricultural soils after afforestation: a meta-analysis. *Global Change Biology*, 16: 439-453. <https://doi.org/10.1111/j.1365-2486.2009.01930.x>
IPCC 2019, Refinement to the 2006 IPCC Guidelines for National Greenhouse Gas Inventories, Volume 4 Agriculture, Forestry and Other Land Use.

Mayer et al. 2020; Tamm Review: Influence of forest management activities on soil organic carbon stocks: A knowledge synthesis, *Forest Ecology and Management*, Volume 466. DOI 10.1016/j.foreco.2020.118127

To Reviewer #1

[General Comment] *The manuscript by Hong, Ding, and colleagues presents data on carbon stocks of trees and soils in plantations of northern China. They use a large data set (163 control sites and 614 tree plantations) to show that carbon sequestration depends on tree species, plantation age, and soil nitrogen. I believe that their data has potential to help clarify a limited, but important, set of questions about the carbon and nitrogen dynamics of plantations overtime. Unfortunately, this manuscript has several limitations that prevent it from improving our knowledge of how tree planting may contribute to Natural Climate Solutions, which seems to be the overarching aim stated in the abstract and introduction.*

[Response] Thanks for your encouraging comments and constructive suggestions, which are very helpful for improving the manuscript. We have carefully revised the manuscript following your suggestions, and made the following major changes:

First, we carefully revised the abstract and introduction to clarify that the primary objective of this study is more about the asymmetric carbon dynamics of biomass and soil after forestation regulated by soil nitrogen, rather than how tree planting may contribute to natural climate solutions.

Second, in the revised manuscript, we reworded ‘afforestation’ into ‘forestation’ throughout the manuscript to better reflect the inclusion of both afforestation and reforestation in our study. We also clarified that forestation is not panacea to climate change and carbon sequestration is just one of many ecological and social aspects to consider.

Third, we added more information in the Methods section and revised our data analysis following your suggestions. We used linear mixed model in place of simple regression, where the site number was added in the model as a random factor to overcome the issue of data non-independence. We also added more information of our laboratory work to clearly clarify how soil properties were measured and calculated. The original vegetation and land use type was involved in data analysis and the results were added in the revised version of manuscript. In addition, we used harmonizing vegetation-specific maps of both above and belowground

biomass to extract biomass in control groups and accordingly revised the results. All these adjustments make our study more solid and convincing.

Finally, we also carefully revised the manuscript following other comments and suggestions raised by the reviewers. Please find point-to-point responses to your comments below.

[Comment 1] *Firstly, I am concerned by the framing of afforestation as a “nature-based solution.” The first sentence of the abstract references the seminar paper on Natural Climate Solutions by Griscom et al. (2018). But that paper focuses on reforestation NOT afforestation. A growing body of literature that demonstrates that afforestation via exotic plantations is problematic both socially and ecologically (see Fleischmann et al. 2020, BioScience) and will not help limit global warming to meet the Paris goals (see Lewis et al. 2019, Nature). These problems in China were also recognized by Cao et al. (2011, Earth Science Reviews), who point out that ecological inappropriate tree planting does more environmental damage than good. All of this is not to say that we should not study afforestation via monoculture tree plantations, but that the framing should clarify that carbon sequestration is just one of many ecological and social aspects to consider.*

[Response] Thanks for your valuable comments. We appreciate your concerns about the framing of forestation as a nature-based climate solution, and made several revisions to the manuscript to address these concerns:

First, we have revised the manuscript to clarify that our study covers both afforestation and reforestation, and used the term "forestation" instead of "afforestation" throughout the paper.

Second, we deleted the frame of nature-based climate solution in abstract and introduction, and clarified that carbon sequestration is just one of many ecological and social aspects of forestation. Accordingly, we discussed the implications of our study for scientific forestation design to maximize carbon benefit and reduce carbon loss (see Line 203-222).

Third, we provided additional discussions to clarify that forestation is not panacea to climate change (Lewis et al., 2019). Well-planned forestation will make ecological and economic

benefits (Fang et al., 2014; Ouyang et al., 2016), while simple and unreasonable forestation may also bring both environment and social problems (Cao et al., 2011; Fleischmann et al., 2020). These contents have been added in Discussion part as: “Furthermore, forestation is not panacea to climate change and carbon sequestration is just one of many ecological and social aspects of forestation³³. It has been found that simple and unreasonable forestation may also bring both environment and social problems^{34, 35}. Therefore, collectively, a sustainable forestation carbon sink requires integrated planning and scientific decision making (to plant or not to plant), accounting for local climates and conditions (where to plant), temporal dynamics of forestation-induced carbon sink (when to plant) and careful species choice and management (how to plant), which are all extremely important if China is to achieve its ‘carbon neutrality’ target.”(Line 222-229).

Collectively, all above revisions make our manuscript more focus on carbon sequestration, especially the asymmetric carbon dynamics of biomass and soil after forestation, than how forestation may contribute to natural climate solutions.

Cao, S. et al. Excessive reliance on afforestation in China's arid and semi-arid regions: Lessons in ecological restoration. *Earth-Science Reviews* **104** 240–245 (2011).

Fleischmann, F. et al. Pitfalls of tree planting show why we need people-centered natural climate solutions. *BioScience* **70**: 947-950 (2020).

Fang, J. et al. Forest biomass carbon sinks in East Asia, with special reference to the relative contributions of forest expansion and forest growth. *Global Change Biology* **20**, 2019–2030 (2014).

Lewis, S. et al. Regenerate natural forests to store carbon. *Nature*, 568, 25-28 (2019).

Ouyang, Z. et al. Improvements in ecosystem services from investments in natural capital. *Science* **352**, 1455-1459 (2016).

[Comment 2] *Secondly, the statistical methods overstate the level of replication. There 614 plantation plots but only 163 control plots, making it so that “several afforested plots (1-26)” (L 309-311) are associated with each control. The statistical methods make it appear that you*

consider the plantation-control pairs to be the unit of replication, with an n=614. This is incorrect because plantation plots located near each control plot are not independent. It would be more appropriate to use the n=163 sites (with 1 control plot per site) as the unit of replication. One solution is perform linear mixed models (rather than simple regressions) and to treat site as a random factor in linear.

[Response] Many thanks for your valuable comments and suggestions, which have greatly improved the quality of our manuscript. Our sampling design ensured that each forested plot was paired with a corresponding control plot that shared similar climate and topography, and the same original vegetation and land use type and soil type. This design allowed us to use the difference between each pair of forested and control plot to assess the impacts of forestation. In some investigated areas, different tree species were planted or the same tree species were planted in different years, resulting in several forested plots with different tree species or stand age sharing the same control plot. Such sampling design is realistic and makes our study more comprehensive, allowing us to identify the different effects of different tree species or age on SOC and biomass under the same conditions.

As the reviewer mentioned, this sampling design make the data not completely independent. To address this issue, we followed your suggestion and used liner mixed model in place of a simple regression, incorporating the site number as a random factor in the data analysis (Fig. R1-Rev1, Fig. R2-Rev1, i.e. Fig. 1 and 3 in the manuscript). This revision did not change our major results but make them more convincing.

We have revised the methods (Line 420-422), figures and results in the manuscript (Fig. 1 and Fig. 3 and corresponding contents) accordingly. In addition, we revised the Abstract section by changing ‘614 control-afforested plot pairs’ into ‘163 control plots and 614 forested plots’ to avoid any potential misunderstanding.

Fig. R1-Rev1 Changes in biomass density ($\Delta(\text{biomass density})$) induced by forestation. a-e Relationships between $\Delta(\text{biomass density})$ and stand age for *P. koraiensis*, *P. sylvestris* var. *mongolica*, *P. tabuliformis*, *L. gmelinii* and *Populus* spp.. The solid lines in panels a-e are the results of **linear mixed model**, while the dashed lines mark the 95% confidence intervals. **f.** The averaged values of $\Delta(\text{biomass density})$ for different planted tree species and overall. Error bars indicate standard errors. The numbers on top of the bars indicate the sample size of each group. **g.** The spatial distribution of $\Delta(\text{biomass density})$ derived from upscaling via model tree ensemble (MTE). **h.** The spatial distribution of the ratio of $\Delta\text{SOCD}/\Delta(\text{biomass density})$. The resolution of the data in **g** and **h** is 1 km. The inset pie chart shows the percentage of each group in the data.

Fig. R2-Rev1 Dependency of changes in carbon densities induced by forestation on background soil total nitrogen density (STND). **a**, Relationship between changes in carbon densities of biomass and SOC with STND. **b**, Relationship between changes in total organic carbon density and STND. Lines in **a** and **b** were fitted based on **linear mixed model**. **c**, Comparison of changes in carbon densities in groups with different STND values. Independent sample t-tests with correction for false discovery rates were conducted to compare the data of each group with 0. *, ** and *** indicate that the null hypothesis can be rejected at $p < 0.05$, 0.01 and 0.001, respectively. Error bars indicate standard errors. This figure is based on field sampling data at plot level. **d**, Trade-off between biomass and soil carbon dynamics among tree species. Increase rate of $\Delta(\text{Biomass density})$ with stand age refers to the slope between $\Delta(\text{Biomass density})$ and stand age (see Fig. 1a-1e). Change rate of ΔSOC with STND refers to the regression slope between ΔSOC and STND.

[Comment 3] Thirdly, the laboratory methods likely overestimate soil bulk density due to inclusion of water weight. Estimation of soil carbon stocks require accurate measurement of

soil bulk density. It is standard to dry soils at 105 C to vaporize water and obtain a dry weight. Based on the methods (L355), soils were simply “air-dried to a constant weight in a ventilated room”. Use of a non-standard method requires explanation.

[Response] We are sorry for omitting such important information. During the field sampling, we collected two identical cutting rings of soils at each depth. One was later oven-dried at 105 °C to determine soil dry weight (SDW) and bulk density (BD) in the laboratory, while the other was air-dried to measure SOC, STN and other properties.

To calculate SOC stock, we used the following equation, which take into account the volume of the cutting ring (V), BD, SDW, contents of SOC (SOCC), and soil depth:

$$\text{SOC}_j = \text{SOCC}_j * \text{BD}_j * P_{s_j} * w_j * 10^2 = \text{SOCC}_j * \frac{\text{SDW}_j}{V} * w_j * 10^2$$

P_s indicates the volume percentage of soil in each cutting ring (100% minus the volume percentage of roots and stones). Note j indicates the j th layer and w indicates the thickness of layer. For example, if $j = 1$, it indicates the first layer (0-5 cm) and $w = 5$ cm. Similarly, if $j = 6$, it indicates the sixth layer (60-100 cm), and $w = 40$ cm. All above information has been provided in the Methods section of revised version following your suggestions (Line 368-371 and Line 393-410).

[Comment 4] *Fourth, the afforestation sites included a wide range of previous land uses, which were not incorporated in models, and could have important implications for carbon dynamics and biodiversity. These included (L316-317), “barren land, cropland, grassland, natural forest, and riparian sand land.” It is incorrect to call the clearing of natural forests with subsequent planting of monoculture tree plantations “afforestation”.*

[Response] Thanks for your comments. We fully acknowledge the importance of considering the impact of the previous land use type on the ecological effects of forestation, as you mentioned. Following your suggestions, we have made some revisions to the manuscript:

First, we added the original vegetation and land use type as a random factor in the linear mixed model to explore its effects on our results. As shown in Table R1-Rev1 and Fig. R3-Rev1, it

only had very slight effect on the results comparing with Fig. R1-Rev1 and Fig. R2-Rev1. Therefore, the original vegetation and land use type did not confound our major findings.

Second, we were unable to include this variable (original vegetation and land use type) in our machine learning model (MTE model) for upscaling estimation of large-scale carbon sink as we did not have access to large-scale data of previous land use types. **Therefore, we attempted to explore the impact of this factor by training MTE models with and without previous land use type involved (Fig. R4-Rev1).** It showed that taking previous land use type into consideration did not substantially improve model performance (it performed better for training data but worse for test data), which indicates the performance of our present model is acceptable. We added these points in the Discussion section “although original vegetation and land use type was found to affect carbon sequestration of forestation, we were unable to include this variable in our MTE model due to the lack of available regional-scale data. However, we conducted further analysis by training MTE models with and without previous land use type involved and found model with vegetation and land use type did not show much better performance (Supplementary Fig. 3), indicating the MTE model used is sufficiently reliable to estimate biomass carbon sink of forestation” (Line 231-237).

Third, to clearly present the impacts of original vegetation and land use type on forestation carbon sequestration, we included new results of carbon changes induced by forestation in different previous land use type (Table. R2-Rev1) and added following statement: “The biomass carbon sequestration of forestation also varied among original vegetation and land use type (Supplementary Table 1). The largest carbon sequestration was observed for forestation on natural forest and cropland, especially when forested with *P. sylvestris* var. *mongholica* on them. By contrast, the smallest carbon sequestration was observed for forestation on grassland.” (Line 98-101).

Finally, we also agree that our study include both afforestation and reforestation. Therefore, we have reworded ‘afforestation’ into ‘forestation’ throughout the manuscript, as it more accurately reflects the inclusion of both practices in our research.

Table R1-Rev1 Slopes between Δ (biomass density) and stand age in linear mixed model (same method with Fig. R1-Rev1). The two rows show the results with and without original vegetation and land use type involved in the model as a random factor.

	P. koraiensi	L. gmelinii	P. sylvestris var. mongholica	P. tabuliformis	Populus spp.
Without OVLUT	0.20	0.37	0.33	0.10	0.18
With OVLUT	0.20	0.37	0.33	0.10	0.17

Table R2-Rev1 Changes in biomass density (kg C m⁻²) induced by forestation with different tree species in different original vegetation and land use type.

	P. koraiensi	L. gmelinii	P. sylvestris var. mongholica	P. tabuliformis	Populus spp.	Overall
Cropland	5.07±0.47 (14)	4.44±0.13 (72)	7.26±0.54 (15)	3.25±0.80 (5)	6.00±0.06 (70)	5.31±0.04 (176)
Barren land	4.09±0.18 (19)	5.68±0.06 (118)	7.20±0.24 (36)	1.26±0.03 (67)	3.74±0.06 (53)	4.41±0.02 (293)
Grassland		1.57±0.49 (8)	3.96±0.37 (6)	1.24 (1)	2.46±0.38 (7)	2.49±0.14 (22)
Natural forest	4.20±0.17 (22)	6.46±0.14 (66)	7.07±2.02 (4)	1.38±0.36 (10)	4.38±1.33 (3)	5.47±0.07 (105)
Riparian sand land					5.24±0.15 (18)	5.24±0.15 (18)
Total	4.38±0.08 (55)	5.41±0.03 (264)	6.89±0.13 (61)	1.40±0.03 (83)	4.92±0.03 (151)	4.80±0.01 (614)

The mean values, standard errors and the sample size (in brackets) are shown in the table. Numbers in bold indicate the changes are significantly different from 0 ($p < 0.05$) based on independent sample *t*-tests.

Fig. R3-Rev1 Test of results in Fig. R2-Rev1. Lines in **a** and **b** were fitted based on linear mixed model. Comparing with Fig. R2-Rev1, the original vegetation and land use type was involved in the model as a random factor.

Fig. R4-Rev1 The performance of model tree ensemble (MTE) without (a) and with (b) original vegetation and land use type involved in estimating changes in biomass density induced by forestation.

[Comment 5] *Fifth, biomass was measured by the authors in the plantation plots, but not in the control plots. This is not only a methodological inconsistency, but assumptions about biomass in the control plots (and corresponding error) calls into question the ΔC estimates for*

the plantations. Specifically (L347-348): “Biomass density in cropland, barren land and riparian sand land was set to be 0.” This is an underestimate of existing biomass (particularly below ground) on these sites. For “natural forest control sites” (L348-350) the authors did not measure trees, but rather used above ground biomass estimates from a remote sensing data set and then multiplied by a constant (1.24). Finally (L350-352), for grassland control sites (which certainly have the majority of biomass below ground) they use a MODIS (remote sensing) product that estimates net primary productivity. But this is a spatial mismatch, because the MODIS resolution is 500 m x 500 m whereas the control and plantation plots are 20 m x 20 m. Furthermore, there is insufficient description in the methods to understand how the MODIS estimates are used to calculate the grassland carbon stocks. All of this is concerning because the control plots are the basis for calculating ΔC biomass, which is key to the aims of the study.

[Response] Many thanks for your valuable comments and suggestions. We would like to apologize for not conducting biomass surveys in areas such as riparian sand land due to the sparse plant growth and the large amount of fieldwork involved. We agree with the reviewer that the previous assumptions of control-group biomass could bring some uncertainties in our results. To improve the calculation of ΔC biomass, we used the harmonizing vegetation-specific maps of both above and belowground biomass to extract biomass density in our control groups. The used dataset (Spawn et al., 2020) comprises both above and belowground biomass for different vegetation types (e.g. cropland, grassland, natural forest al.) in 2010, close to our sampling years (2012 and 2013). The finer resolution of this dataset (300 m) can also better match the size of our sampling plots than previous version (500 m). During the field investigation, we tried our best to ensure that our sampling plots were as representative as possible for the local environment, which can minimize the uncertainties arising from the mismatch in the spatial resolution (20 m and 300 m). Therefore, this dataset is more appropriate to our study and can sufficiently decrease the uncertainty of our estimation.

Using the new dataset, we reconducted all data analyses of our study. The estimated biomass carbon sequestration is 678.25 ± 37.98 Tg C (Table R3-Rev1), lower than previous estimation (735.69 ± 33.41 Tg C). It indicates that our previous assumption underestimated biomass in control groups and thus overestimated forestation carbon sink, as the reviewer suggested.

Nevertheless, this adjustment did not change our major finding that soil nitrogen regulates the asymmetric responses of biomass and SOC to forestation (Fig. R5-Rev1). Therefore, the changes in datasets for control groups did not confound our study but made it more solid.

We have correspondingly revised the manuscript based on the new results. Thank you again for your constructive comments and suggestions.

Table R3-Rev1 Results of the upscaling estimation of forestation induced carbon sequestration.

Change in C density (kg C m ⁻²)			
	Δ (Biomass density)	Δ SOCD	Δ TOCD
P. koraiensis	8.98 (0.53)	4.20 (0.09)	13.18 (0.62)
L. gmelinii	7.54 (0.31)	3.78 (0.08)	11.32 (0.39)
P. sylvestris var. mongholica	3.25 (0.52)	0.95 (0.28)	4.20 (0.80)
P. tabuliformis	0.54 (0.10)	1.28 (0.05)	1.82 (0.15)
Populus spp.	4.86 (0.20)	0.73 (0.11)	5.59 (0.31)
Other	6.07 (0.30)	0.91 (0.03)	6.98 (0.33)
Average	5.51 (0.31)	1.91 (0.08)	7.42 (0.39)
Change in Total C (Tg C)			
	Biomass	SOC	TOC
P. koraiensis	22.29 (1.34)	11.26 (0.17)	33.55 (1.51)
L. gmelinii	236.97 (13.93)	109.87 (1.80)	346.84 (15.73)
P. sylvestris var. mongholica	14.96 (1.95)	4.05 (0.21)	19.01 (2.16)
P. tabuliformis	8.61 (1.52)	17.70 (0.58)	26.31 (2.10)
Populus spp.	227.36 (7.93)	59.89 (4.30)	287.25 (12.23)
Other	168.05 (11.30)	32.17 (1.94)	200.22 (13.24)
Total	678.25 (37.98)	234.94 (9.60)	913.19 (47.58)

Numbers in the brackets indicate the standard errors for multiple simulations. Note that the average changes in C density were calculated using the total C change divided by the area of planted forests.

Fig. R5-Rev1 Dependency of changes in carbon densities induced by forestation on background soil total nitrogen density (STND). **a.** Relationship between changes in carbon densities of biomass and SOC with STND. **b.** The relationship between changes in total organic carbon density and STND. Lines in **a** and **b** were fitted based on linear mixed model. **c.** Comparison of changes in carbon densities in groups with different STND values. Independent sample *t*-tests with correction for false discovery rates were conducted to compare the data of each group with 0. *, ** and *** indicate that the null hypothesis can be rejected at $p < 0.05$, 0.01 and 0.001, respectively. Error bars indicate standard errors. This figure is based on field sampling data at plot level. **d** Trade-off between biomass and soil carbon dynamics among tree species. Increase rate of $\Delta(\text{Biomass density})$ with stand age refers to the slope between $\Delta(\text{Biomass density})$ and stand age (see Fig. 1a-1e). Change rate of ΔSOC with STND refers to the regression slope between ΔSOC and STND.

Spawn, S. et al. Harmonized global maps of above and belowground biomass carbon density in the year 2010. *Scientific Data* 7:112 (2020).

[Comment 6] *In closing, I hope that the authors can use these critiques to revise and focus their analyses on a set of refined hypotheses for which their data is better suited. I trust that with such a comprehensive data set on plantation carbon stocks and Nitrogen that such a revision will focus on C and N dynamics over time, and less so on climate mitigation via C sequestration.*

[Response] Thanks again for your encouraging words and constructive suggestions, which helped us a lot to generate a better manuscript. Following your suggestions, we discarded our assumptions on biomass in control groups and used a more comprehensive dataset with higher resolution (vegetation-specific maps of both above and belowground biomass). This dataset is more suitable for the size of our sampling plots and thus can help improve the accuracy of our analyses. In addition, we carefully revised the abstract, introduction and discussions following your suggestions to focus our study on C and N dynamics over time (especially how soil N regulates the asymmetric dynamics of biomass and soil C after forestation), rather than climate mitigation via carbon sequestration. These revisions made our study more solid and convincing. Thank you again for your big help and we hope the revised version of manuscript will satisfy you.

To Reviewer #2

[General Comment] *This study uses a remarkable dataset where soil organic carbon and plant biomass have been assessed over 163 study sites of afforested and control plots over a large region in China. The study has several shortcomings that prevent me from making a positive recommendation.*

[Response] Thank you for taking the time to review our manuscript and providing valuable comments, which have been immensely helpful in improving the quality of our manuscript. We made significant revisions to address the shortcomings that you identified, and the major changes included are summarized below:

First, we clearly expressed our sampling design and added more information about the control treatment. Especially, we clarified that we focused more on forestation versus non-forestation comparisons than before-and-after comparisons since the control plot could not perfectly represent the initial conditions.

Second, we provided additional detailed information of laboratory work and reconducted data processing following your suggestions. Specifically, we used linear mixed model in place of simple regression in data analysis. The site number was added in the model as a random factor to overcome the issue of non-independence.

Thirdly, we carefully adjusted the introduction and discussion sections to avoid to be speculative. We revised our assumption and provided additional discussion on the causes of SOC dynamics after forestation, and only had priming effect as a potential explanatory mechanism of the trade-off between biomass and SOC.

Finally, we clearly clarified the shortage of Earth System model and large-scale C estimation, and discussed the implications of our results (trade-off between biomass and SOC after forestation along soil N gradient) for improving large-scale C modeling and estimations.

Detail responses are provided below.

[Comment 1] *-Lack of information on the control treatment. Although an effort was made so that both treatments (afforested and control) shared the same initial conditions at the time of plantation, it is unclear how and whether the control plots were managed or not. This could have important implications for the interpretation of the results and this information is lacking. In particular, I understand that plantations of different ages were sampled once; i.e. there is no re-sampling as the plantations are aging. Therefore, the control may not reflect the initial conditions. This aspect needs to be clarified because the study is based on the assumption that the control represents plantation conditions at time zero.*

[Response] Thanks for your comments and suggestions. The control plots in our study include five types of vegetation and land use (cropland, barren land, grassland, natural forest and riparian sand land). During the investigation, we took efforts (combining the data of local forestry bureau and field survey) to make sure that corresponding control and forested plots shared the same original conditions. Except cropland, other four types were not managed. Above information has been added to the revised manuscript (Line 345-351).

We acknowledge that the control plot can represent the environmental conditions without forestation treatment, but it may not necessarily represent the plantation conditions at time zero. Therefore, we did not rely on the comparison of C stocks before and after forestation, which can also be influenced by climate change, nitrogen deposition and myriad other factors. Instead, we focused on comparing C stocks under forestation versus no forestation, all else equal, which indicate the cost/benefit of forestation. We have clarified this point in the revised manuscript: “Each control-forested pair, consisting of a forested plot and its corresponding control plot was utilized to provide a good assessment of the impact of forestation. It is also noteworthy that we do not rely on before-and-after comparisons since the control plot could not accurately represent the initial conditions. Instead, we focused on the differences in C stocks between forested versus non-forested areas, all else equal. This approach enables us to evaluate the cost/benefit of forestation accurately.” (Line 351-356).

[Comment 2] *-Methodology:*

1- It is unclear whether residues, litter, or organic layers are included or not in the SOM assessment.

[Response] Residues, litter, organic layers are not included in the SOM assessment. We have added this information in the revised manuscript: “Note that we collected two cutting rings of soils at each depth, both of which were identical. One of rings was oven-dried while the other was air-dried. It should be noted that residues, litter, organic layers were not included during the soil sampling process” (Line 368-371).

[Comment 3] *2- There is no mention that soil bulk density, a crucial element for estimating SOM stocks was assessed. There is also no information on the exclusion of coarse fragments (greater than 2mm). Although a cutting ring is mentioned in l.360; there is no mention of how rock fragments were dealt with.*

[Response] We apologize for omitting such important information. We have made significant revisions to address the concerns you raised.

Regarding soil bulk density, we collected two identical cutting rings of soils at each depth during the field sampling. One of them was oven-dried at 105 °C to constant weight to determine the soil dry weight (SDW) and bulk density (BD) in the laboratory, while the other air-dried for SOC, STN and other properties. We also excluded coarse fragments (greater than 2mm) by passing the sieved soils through 2-mm sieves.

To calculate SOC stock, we used the following equation, which take into account the volume of the cutting ring (V), BD, soil dry weight, contents of SOC, and soil depth:

$$\text{SOC}_j = \text{SOC}_j * \text{BD}_j * \text{Ps}_j * w_j * 10^2 = \text{SOC}_j * \frac{\text{SDW}_j}{V} * w_j * 10^2$$

Ps indicates the volume percentage of soil in each cutting ring (100% minus the volume percentage of roots and stones). Note j indicates the jth layer and w indicates the thickness of layer. For example, if j = 1, it indicates the first layer (0-5 cm) and w = 5 cm. Similarly, if j = 6, it indicates the sixth layer (60-100 cm), and w = 40 cm. All above information has been provided in the revised version following your suggestions (Line 368-371 and Line 393-410).

[Comment 4] -*Statistical treatment:*

There are on average six afforested plots that are compared to a single control plot. How was this considered in the statistical analysis? Also, the statement that there are 614 control-afforested plot pairs appears to be misleading as the number of pairs should be 163 (l. 309).

[Response] Thanks for your comments. In this study, each forested plot and its corresponding control plot are well paired (close distance, the same original vegetation and land use type, the same topography and soil type), so the difference between each pair of forested and control plot can well indicate the impacts of forestation. In some areas, different tree species (or same tree species but planted in different years) were planted, so several forested plots with different planted tree species or stand age may share the same control plot, which represent the different effects of different tree species (or age) on SOC and biomass in the same original condition. Such design makes our study more comprehensive and reliable, but would also make the data non-independent. To solve this problem, following the suggestions of you and Reviewer #1, we performed liner mixed model in place of simple regression and included the site number as a random variable (Fig. R1-Rev2 and R2-Rev2). You will see this revision did not change our major results but make them more convincing and solid. We accordingly revised the methods, figures and corresponding contents.

Furthermore, following the suggestions of reviewers, we revised ‘614 control-forested plot pairs’ in the manuscript as ‘163 control plots and 614 forested plots’ to avoid potential misunderstanding.

Fig. R1-Rev2 Changes in biomass density ($\Delta(\text{biomass density})$) induced by forestation. a-e Relationships between $\Delta(\text{biomass density})$ and stand age for *P. koraiensis*, *P. sylvestris* var. *mongolica*, *P. tabuliformis*, *L. gmelinii* and *Populus* spp.. The solid lines in panels a-e are the results of **linear mixed model**, while the dashed lines mark the 95% confidence interval of the regressions. **f.** The averaged values of $\Delta(\text{biomass density})$ for different planted tree species and overall. Error bars indicate standard errors. The numbers on top of the bars indicate the sample size of each group. **g.** The spatial distribution of $\Delta(\text{biomass density})$ derived from upscaling via model tree ensemble (MTE). **h.** The spatial distribution of the ratio of $\Delta\text{SOC}/\Delta(\text{biomass density})$. The resolution of the data in **g** and **h** is 1 km. The inset pie chart shows the percentage of each group in the data.

Fig. R2-Rev2 Dependency of changes in carbon densities induced by forestation on background soil total nitrogen density (STND). **a**, Relationship between changes in carbon densities of biomass and SOC with STND. **b**, Relationship between changes in total organic carbon density and STND. Lines in **a** and **b** were fitted based on linear mixed model. **c**, Comparison of changes in carbon densities in groups with different STND values. Independent sample t-tests with correction for false discovery rates were conducted to compare the data of each group with 0. *, ** and *** indicate that the null hypothesis can be rejected at $p < 0.05$, 0.01 and 0.001, respectively. Error bars indicate standard errors. This figure is based on field sampling data at plot level. **d**, Trade-off between biomass and soil carbon dynamics among tree species. Increase rate of $\Delta(\text{Biomass density})$ with stand age refers to the slope between $\Delta(\text{Biomass density})$ and stand age (see Fig. 1a-1e). Change rate of ΔSOC with STND refers to the regression slope between ΔSOC and STND.

[Comment 5] -Speculative assumptions:

1-In the manuscript, any loss of soil C is interpreted as being the product of priming. Soil organic matter priming is commonly defined as the modification of soil organic matter (SOM) decomposition by plant carbon (C) input. The authors present no evidence that changes in SOM are caused by plant carbon inputs. Plant carbon inputs have not been assessed. Other causes of SOM changes, including disturbance, management, and changes in soil biological or microclimatic conditions have not been assessed and are not discussed. L.78; 207; 249; 254; 264; 272 (strong priming effect).

[Response] We appreciate your valuable comments and suggestions regarding the factors affecting soil organic carbon (SOC) dynamics and the need for a more nuanced discussion of the priming effect, and made the following revisions to address your concerns:

First, we agree with the reviewer that the dynamic of SOC is determined by the balance of C inputs and outputs (Davidson & Janssens, 2006; Binkley & Fisher, 2013), and other factors such as change in soil biological or microclimatic conditions (soil temperature, moisture, pH and et.), disturbance, management would also affect such balance (Laganière et al., 2010; Wang et al., 2011; Peng et al., 2014). We apologize for leaving the impression that we solely focused on the changes in SOC after forestation or interpreting any losses of SOC as the product of priming effect. As you suggested, we have revised the manuscript to clarify our assumptions and discussions on biomass and SOC dynamics, highlighting priming effect as only one potential explanatory mechanism of the trade-off between biomass and SOC: “**Second, the association between biomass density and SOCD changes along the soil nitrogen gradient, implies that the above- and below-ground interactions are regulated by plant nutrient acquisition. In general, a larger biomass can produce more litter and increase carbon input to the soil carbon pool. Thus, a positive relationship between biomass and SOC is widely used in most terrestrial ecosystem models^{14,42}. However, the increase of plant biomass requires a larger nutrient supply, and can stimulate the decomposition of soil organic matter to obtain more nitrogen⁴³. In areas with a large amount of soil organic matter, the input of litter to the SOC pool cannot completely recharge the strong decomposition of SOC (maybe due to the priming effect)⁴⁴, and hence we observed a large decrease of SOC.” (Line 263-271).**

Secondly, we also added a paragraph in Discussion section to provide additional discussion on

the causes of SOC dynamics after forestation, emphasizing the complex interplay between forestation, litter input, nutrient acquisition, soil biological or microclimatic conditions, disturbance, and management “Forestation regulates the dynamics of SOC via affecting both C inputs and outputs^{50, 51}. Besides the litter input and nutrient acquisition (e.g. priming effect), forestation could also regulate SOC dynamics indirectly via changing soil biological or microclimatic conditions (soil temperature, moisture, pH and et.)^{52, 53}. Moreover, disturbance and forest management can also affect the input and output of SOC⁵⁴⁻⁵⁶. These effects make dynamics of SOC after forestation more complicated and leave the interaction between biomass and soil C cycles more uncertain.” (Line 290-295).

Thirdly, we provided more evidence to support our claim that the trade-off between biomass and soil carbon dynamics is related to plant nitrogen acquisition and the priming effect: Many factors could regulate the balance of C inputs and outputs after forestation, but they do not fully explain the observed trade-off between biomass and SOC along soil nitrogen gradient in this study. Nevertheless, such trade-off could be explained by plant nitrogen acquisition and priming effect. Indeed, when comparing the changes in soil pH, soil nitrogen and C:N after forestation along soil nitrogen gradient, we found the change in soil N had similar patterns to soil C, but seemed to be opposite to that of biomass change (Fig. R3-Rev2), indicating that plant nitrogen acquisition may be the major cause of the trade-off between biomass and SOC. Long-term control experiments supported this mechanism. For instance, Castañeda-Gómez et al. (2023) found that doubled aboveground litter additions did not increase soil C for any of the forests studied, likely due to long-term soil priming. Moreover, they also demonstrated that degree of SOM decomposition was higher for sites with higher N availability while lower for N-poor forests. Such results are consistent with our finding that soil nitrogen regulated the asymmetry of carbon sequestrations by plant and soil after forestation. Trade-off between plant soil carbon storage was also found under elevated CO₂ (Terrer et al., 2021). Terrer et al. (2021) synthesized data from 108 eCO₂ experiments and found that the effect of eCO₂ on SOC stocks is best explained by a negative relationship with plant biomass: when plant biomass is strongly stimulated by eCO₂, SOC storage declines; conversely, when biomass is weakly stimulated, SOC storage increases. Consistent with our results, they also demonstrated this trade-off

appears to be related to plant nutrient acquisition, in which plants increase their biomass by mining the soil for nutrients, which decreases SOC storage. Moreover, in tundra, where large amount of SOC are stored, it was found that high plant activity during the growing season stimulates the decomposition of soil organic matter (Hartley et al., 2012). Hartley et al. (2012) further demonstrated that such a response, referred to as positive priming, helps explain the low soil C storage in the forest when compared with the tundra. Collectively, although many factors could regulate SOC dynamics after afforestation, the observed trade-off between biomass and SOC is closely linked to plant nitrogen acquisition, and priming effect is a potential mechanism. All these evidences have been added and discussed in Line 271-279.

Fig. R3-Rev2 Changes in soil pH, soil total nitrogen density (STND) and soil C:N induced by forestation along background STN gradient. Independent sample t-tests with correction for false discovery rates were conducted to compare the data of each group with 0. *, ** and *** indicate that the null hypothesis can be rejected at $p < 0.05$, 0.01 and 0.001, respectively. Error bars indicate standard errors.

- Castañeda-Gómez, L. et al. Soil organic matter molecular composition with long-term detrital alterations is controlled by site-specific forest properties. *Glob Change Biol.* **29**:243–259 (2023).
- Davidson, E. & Janssens, I. Temperature sensitivity of soil carbon decomposition and feedbacks to climate change. *Nature* **440**, 165–173 (2006).
- Hartley, I. et al. A potential loss of carbon associated with greater plant growth in the European Arctic. *Nature Climate Change* **2**, 875-879 (2012).
- Laganière, J. et al. Carbon accumulation in agricultural soils after afforestation: a meta-analysis. *Glob Change Biol.* **16**: 439-453 (2010).
- Peng, S. et al. Afforestation in China cools local land surface temperature. *Proc. Natl Acad. Sci. USA* **111**, 2915–2919 (2014).
- Terrer, C. et al. A trade-off between plant and soil carbon storage under elevated CO₂. *Nature* **591**, 599-603 (2021).
- Wang, W. et al. Changes in soil organic carbon, nitrogen, pH and bulk density with the development of larch (*Larix gmelinii*) plantations in China. *Glob. Change Biol.* **17**, 2657–2676 (2011).

[Comment 6] *2-The authors have frequently referred to the assumption that many studies, assessments, and modeling use a direct relationship between biomass accumulation and changes in SOM. I do not believe that this is true, contrary to the authors' claim. Some model results would show such a correlation; but it is generally accepted that while plant biomass accumulates after afforestation, changes in SOM are frequently not apparent and can be negative or positive depending on several parameters related to previous land use and soil properties. The common understanding of the factors that are known to impact SOM accumulation with afforestation is not discussed. For example, a well-cited meta-analysis (Laganière et al. 2010) has identified major drivers of SOM accumulation following afforestation including land use and soil types. Also, the IPCC (2019), in its updated methodology based on an extensive literature review, does not refer to a relationship between productivity and SOM accumulation (IPCC). See also Mayer (2020).*

Cited literature:

LAGANIÈRE, et al. (2010), *Carbon accumulation in agricultural soils after afforestation: a meta-analysis*. *Global Change Biology*, 16: 439-453. <https://doi.org/10.1111/j.1365-2486.2009.01930.x>

IPCC 2019, *Refinement to the 2006 IPCC Guidelines for National Greenhouse Gas Inventories, Volume 4 Agriculture, Forestry and Other Land Use*.

Mayer et al. 2020; *Tamm Review: Influence of forest management activities on soil organic carbon stocks: A knowledge synthesis, Forest Ecology and Management, Volume 466*. DOI 10.1016/j.foreco.2020.118127

[Response] Thanks for your insightful comments and suggestions. We would like to express our sincere apologies for any misunderstandings that may have arisen from our previous statement regarding the dynamics of SOC after forestation. We fully acknowledge and agree with your point that although forestation may result in an increase in plant biomass, the changes in soil organic carbon can be complex and contingent on several factors, including original vegetation and land use type etc, which have been examined through sampling- and meta-based studies (Laganière et al., 2010; Wang et al., 2011; Mayer et al., 2020).

Despite these complexities, we recognize that the treatment of biomass accumulation and soil carbon dynamics in large-scale assessments and modelling of ecosystem carbon dynamics is often oversimplified. For example, some studies use fixed ratios of soil C and vegetation biomass to estimate soil C stocks and changes due to the lack of inventory data in certain regions and countries, as was the case in a comprehensive estimation of global forest carbon sink (Pan et al., 2011), and in other studies on terrestrial ecosystem C dynamics (e.g. Shvidenko & Nilsson, 2003; Liu et al., 2015). In addition, Earth system models (ESMs) generally produce a direct relationship between biomass accumulation and changes in SOC, which may oversimplify the complex relationships between these variables (Todd-Brown et al., 2013; Tian et al., 2015). This oversimplification can potentially lead to inaccurate estimations and models of carbon dynamics, emphasizing the need for more comprehensive and nuanced approaches to incorporate the complexities of SOC dynamics in large-scale carbon estimation and modelling.

In revised manuscript, we have added more discussion to explore the complexity of soil carbon

dynamics and potential influencing factors after forestation, and have cited several references you mentioned to further support our discussion: “Forestation regulates the dynamics of SOC via affecting both C inputs and outputs^{50, 51}. Besides the litter input and nutrient acquisition (e.g. priming effect), forestation could also regulate SOC dynamics indirectly via changing soil biological or microclimatic conditions (soil temperature, moisture, pH and et.)^{52, 53}. Moreover, disturbance and forest management can also affect the input and output of SOC⁵⁴⁻⁵⁶. These effects make dynamics of SOC after forestation more complicated and leave the interaction between biomass and soil C cycles more uncertain.” (Line 290-295).

Moreover, we also clearly clarified current knowledge gap in large-scale SOC estimation and modelling, and further specified the implications of our study in filling this knowledge gap “At present, Earth system models (ESMs) generally produce a strong response of SOC to an increase in C input (i.e. NPP)^{14, 42}. However, our results suggest that the dynamics of plant C and SOC are not proportionally synergistic and there may be a trade-off between them due to nutrient competition. Such nutrient competition mechanisms have also been observed in tundra forest⁴⁵, and further confirmed by a synthesized study focusing on CO₂ fertilization²⁰. Moreover, both modeling⁴² and field experiment⁵⁷ studies have found that increasing N input can enhance soil C sequestration. Given that ESMs are currently inadequate for modeling C and N couples⁵⁸, a better parameterization and description of C and N interaction schemes is likely to reduce model uncertainties in SOC dynamic simulation. Integrated studies, combining data from manipulative field experiments and large-scale sampling of soil C and N dynamics may yield some new insights, but we suggest that ecological theories of nutrient acquisition can also help to develop and refine the soil C-N schemes used in ESMs.” (Line 297-308).

Laganière, J. et al. Carbon accumulation in agricultural soils after afforestation: a meta-analysis. *Glob Change Biol.* **16**: 439-453 (2010).

Liu, Y. et al. Recent reversal in loss of global terrestrial biomass. *Nature Climate Change* **5**(5):470-474 (2015).

Mayer et al. Tamm Review: Influence of forest management activities on soil organic carbon stocks: A knowledge synthesis, *Forest Ecology and Management* **466**, (2020).

- Pan, Y. et al. A large and persistent carbon sink in the world's forests. *Science* **333**, 988–993 (2011).
- Shvidenko, A. & Nilsson, S. A synthesis of the impact of Russian forests on the global carbon budget for 1961–1998. *Tellus* **55B**, 391–415 (2003).
- Tian, H. et al. Global patterns and controls of soil organic carbon dynamics as simulated by multiple terrestrial biosphere models: current status and future directions. *Glob. Biogeochem. Cycles* **29** 775–792 (2015).
- Todd-Brown, K. E. O. et al. Causes of variation in soil carbon simulations from CMIP5 Earth system models and comparison with observations. *Biogeosciences* **10**, 1717–1736 (2013).
- Wang, W. et al. Changes in soil organic carbon, nitrogen, pH and bulk density with the development of larch (*Larix gmelinii*) plantations in China. *Glob. Change Biol.* **17**, 2657–2676 (2011).

Reviewer #2 (Remarks to the Author):

The authors' response and the revised version of the manuscript have clarified most of the issues that I had raised concerning the methodology and the interpretation of the results.

I appreciate the more balanced discussion on the potential contribution of a priming effect to explain the relationship between soil C loss in high N soils.

The large dataset and the large range of environmental conditions that are covered make this study very valuable. The potential explanation for the results, specifically the importance of priming to explain the decline in soil C in soil with high N is presented with more nuances and is certainly an interesting hypothesis that can stimulate further research. Please consider the following comments for this new version:

1-The study covers a large range of mean annual temperature MAT (-4 to +14). Along this gradient, the availability of water to plants and to soil organisms is much more related to aridity than to precipitation per se (i.e.: at a same precipitation level, a cold site would be much wetter than a warm site). Using an aridity index such as the FAO Penman-Monteith Reference Evapotranspiration (ET₀) equation (Zomer, Xu & Trabucco 2022) instead of precipitation, especially in Fig. 2 (a,b,c,d) could help to better elucidate the role of soil water availability on these processes versus that of temperature.

For example, this statement (L. 210: Specifically, forestation in humid (cold) regions sequesters carbon more effectively than in dry (warm) regions (Fig. 2).) is not obvious when looking at Fig.2 but would probably be more convincing with an aridity index.

2-The information presented in the response to reviewers, specifically Fig.R3-Rev2 is pertinent to the interpretation of the results, and I would suggest having them included in the study or at the very least in the supplementary material. Also I would be curious to know if STND (soil total nitrogen density) that is used to qualify N availability is also correlated with soil C:N a well know indicator of N availability.

Minor comments:

l.212: the formulations nitrogen-moderate regions and nitrogen-rich regions are awkward.

Suggestion: in areas with moderate/high soil nitrogen levels

l. 277: no s in soil organic matter.

Ref:

Zomer, R.J., Xu, J. & Trabucco, A. Version 3 of the Global Aridity Index and Potential Evapotranspiration Database. Sci Data 9, 409 (2022). <https://doi.org/10.1038/s41597-022-01493-1>

To Reviewer #2

[General Comment] *The authors' response and the revised version of the manuscript have clarified most of the issues that I had raised concerning the methodology and the interpretation of the results.*

I appreciate the more balanced discussion on the potential contribution of a priming effect to explain the relationship between soil C loss in high N soils.

The large dataset and the large range of environmental conditions that are covered make this study very valuable. The potential explanation for the results, specifically the importance of priming to explain the decline in soil C in soil with high N is presented with more nuances and is certainly an interesting hypothesis that can stimulate further research. Please consider the following comments for this new version:

[Response] Thanks for your encouraging comments and constructive suggestions, which are very helpful for improving the manuscript. We are glad that you find we have well addressed most comments and suggestions. Following your valuable suggestions, we conducted additional analyses and carefully revised the manuscript. Thank you again for your comments and constructive suggestions in first- and this-time reviews, which truly helped us to generate a much better manuscript. Please find point-to-point responses to your comments below.

[Comment 1] *1-The study covers a large range of mean annual temperature MAT (-4 to +14). Along this gradient, the availability of water to plants and to soil organisms is much more related to aridity than to precipitation per se (i.e.: at a same precipitation level, a cold site would be much wetter than a warm site). Using an aridity index such as the FAO Penman-Monteith Reference Evapotranspiration (ET₀) equation (Zomer, Xu & Trabucco 2022) instead of precipitation, especially in Fig. 2 (a,b,c,d) could help to better elucidate the role of soil water availability on these processes versus that of temperature.*

For example, this statement (L. 210: Specifically, forestation in humid (cold) regions sequesters carbon more effectively than in dry (warm) regions (Fig. 2).) is not obvious when looking at Fig.2 but would probably be more convincing with an aridity index.

Ref: Zomer, R.J., Xu, J. & Trabucco, A. Version 3 of the Global Aridity Index and Potential Evapotranspiration Database. *Sci Data* 9, 409 (2022). <https://doi.org/10.1038/s41597-022-01493-1>

[Response] Many thanks for your constructive suggestions. We agree with the reviewer that the aridity index (AI) could better indicate water availability than precipitation and further strengthen our results. Therefore, following your suggestions, we conducted additional analyses using the AI data recommended by the reviewer. As shown in Fig. R1-Rev2, consistent results were observed with Fig. 2 in the manuscript (based on precipitation), and they both indicate that local climate regulates carbon sequestration induced by forestation. These results have been added in the revised manuscript as “Consistent results were observed when aridity index was used in place of MAP (Supplementary Fig. 3).” (Line 150-151).

Figure R1-Rev2. Divergent responses of biomass and SOC densities to forestation ($\Delta(\text{biomass density})$ and ΔSOC) along climate gradients. a-d. The distribution of $\Delta(\text{biomass density})$, ΔSOC , ΔTOCD (i.e. $\Delta(\text{biomass density})+\Delta\text{SOC}$), and $\Delta\text{SOC}/\Delta(\text{biomass density})$ in a two-dimension space of mean annual temperature (MAT) and aridity index (AI). The mean values for each interval, derived from the output of machine learning, are shown. The top line chart in panel **a** indicates the variation of $\Delta(\text{biomass density})$ along the AI gradient, while the chart on the right-hand side indicates the variation of $\Delta(\text{biomass density})$ with MAT. Mean values for each interval were used to generate the lines. The line charts in panel **b-d** were created similarly.

[Comment 2] 2-The information presented in the response to reviewers, specifically Fig.R3-Rev2 is pertinent to the interpretation of the results, and I would suggest having them included

in the study or at the very least in the supplementary material. Also I would be curious to know if STND (soil total nitrogen density) that is used to qualify N availability is also correlated with soil C:N a well know indicator of N availability.

[Response] Thanks for your valuable suggestions. Following your suggestions, we have included the previous Fig. R3-Rev2 in the supplementary materials to support the interpretation of the results (Line 272-274: “Changes in soil nitrogen after forestation along the nitrogen gradient support this mechanism (Supplementary Fig. 5), where we found significant decrease of soil nitrogen after forestation in areas rich in soil organic matter”). We agree with the reviewer that soil C:N also well indicate nitrogen availability. As soil C:N is directly calculated by SOCD divided by STND, C:N and STND follow a obviously negative relationship (Fig. R2-Rev3). We further tried to explore how soil C:N regulates forestation carbon sequestration in biomass and soils (Fig. R3-Rev2). However, we did not observed any significant relationships between $\Delta(\text{biomass density})$ or ΔSOCD and soil C:N in the linear mixed model.

Figure R2-Rev2. Relationship between soil total nitrogen density (STND) and soil C:N.

Figure R3-Rev2. Relationship between changes in carbon densities of biomass and SOC with soil C:N.

[Comment 3] *l.212: the formulations nitrogen-moderate regions and nitrogen-rich regions are awkward. Suggestion: in areas with moderate/high soil nitrogen levels*

[Response] Thanks for your suggestions. We have adjusted them following your suggestions.

[Comment 4] *l. 277: no s in soil organic matter.*

[Response] We have revised it.